# Monetary Fiscal Contributions to Households and Pension Fund Withdrawals during the COVID-19 Pandemic: An Approximation of Their Impact on Construction Labor Supply in Chile

**Byron J. Idrovo-Aguirre** [1] **and Javier E. Contreras-Reyes** [2,*] 

1    Gerencia de Estudios, Cámara Chilena de la Construcción, Santiago 7560860, Chile; bidrovo@cchc.cl
2    Facultad de Ciencias, Instituto de Estadística, Universidad de Valparaíso, Valparaíso 2360102, Chile
\*    Correspondence: jecontrr@uc.cl; Tel.: +56-(32)-250-8242

**Abstract:** We show statistical evidence that pension fund withdrawals and the Emergency Family Income (EFI) increased the likelihood that a laid off construction worker would reject a proposal for a formal employment contract. This favors the hypothesis that pension fund withdrawals and government subsidies related to the health crisis have, to some extent, contributed to the shortage of formal labor in the construction sector. Based on estimations of the logit model, we found that rejection probability increased with work experience (approximated by the worker's age). For example, the probability of not accepting a formal contract for a highly experienced worker, who withdrew funds from their mandatory private Pension Fund Administrator and received the EFI, increased by 28%. The figure is approximately 2.5 times the rejection probability of a worker with the same experience but without having received this additional income.

**Keywords:** pension funds; emergency family income; policy decision-making; construction activity; logit model; Chile





## 1. Introduction

As in other countries, the COVID-19 pandemic has also produced a great social, political and economic impact in Chile's dense urban areas (Asahi et al. 2021; Bennett 2021). As recently studied, the government established a system of financial pandemic aid for the affected public (Bennett 2021; Fosco and Zurita 2021; Gil et al. 2021; Gozzi et al. 2021; Mena et al. 2021). In particular, the Emergency Family Income (EFI) (Gobierno de Chile 2021) was established in parallel to the ability to withdraw funds from the mandatory privately-managed pension system of the Pension Fund Administrator (PFA) (Behrman et al. 2011; Berstein et al. 2013; Mittelstaedt and Olsen 2003). Crucially, the EFI is a liquidity measure the government implemented for lower-income families (Gozzi et al. 2021), part of the Social Protection Network in response to sanitary provisions like quarantines.

Preliminarily, Modrego et al. (2020) analyzed and predicted the effects of COVID-19 in various Chilean regions. Job losses were estimated at 705,000 on average, 577,000 in the most optimistic case and 870,000 in the worst scenario. Impacts by region were expected to be heterogeneous: Around 1.5% of unemployment in Antofagasta Region but 13.6% in Los Lagos Region. However, the pandemic affected unemployment to a greater extent than expected prior to the official close of 2020. Records of the National Statistics Institute (Instituto Nacional de Estadísticas 2021) revealed that in 2020 more than one million jobs were destroyed nationally because of the pandemic, averaging an unemployment rate close to 11%. This figure contrasts starkly with the 7% unemployment rate of 2019. On the other hand, Coquimbo Region registered the highest unemployment rate in 2020 with 12.4%, followed by Valparaíso Region (11.7%) and the Metropolitan Region (11.3%). The impact of COVID-19 on unemployment is important for the advancement of aid at the regional and national level for social protection and economic stimulus policies. In this

context, the government implemented the universal EFI in 2021, covering more families with monetary contributions to cushion the economic and social impact new confinement measures would cause.

Our analysis focused on the construction sector because of its importance for Chile's labor market. Idrovo-Aguirre and Serey (Idrovo-Aguirre and Serey 2018) showed that the labor factor contribution to the Gross Domestic Product of Construction (GDPC) exceeds 70%, with the rest, less than 30%, coming from the use of the capital factor. In turn, GDPC represents 7% of national GDP. This means construction has generated over US$15 billion a year in added value and created 740,000 jobs a year during the five years prior to the pandemic (2015–2019). The construction sector also accounts for about 64% of the country's total investment of US$42 billion annually from 2015 to 2019. Hence, the sector is predominant in the Chilean economy. The pandemic's impact on the construction market has been as destructive as it was at the national level. Indeed, 146,471 construction jobs were lost, with the Metropolitan, Valparaíso and La Araucanía regions experiencing the highest unemployment. Likewise, GDPC and sector investment fell 14.1% and 11.3% annually, respectively, generating the largest contractions in over 30 years.

So far in 2021, construction activity and investment have not recovered to 2019 levels and several companies still report difficulties in finding labor (Banco Central de Chile 2021a; Cámara Chilena de la Construcción 2021). This situation, in part, is limiting the process of reactivation of the construction industry. The objective of this study was to test the hypothesis that the pension fund withdrawals and the EFI have contributed partially to the shortage of formal construction labor (Contreras-Reyes and Idrovo-Aguirre 2020; Idrovo-Aguirre et al. 2021; Idrovo-Aguirre and Contreras-Reyes 2019; Idrovo-Aguirre and Contreras-Reyes 2021a, 2021b). We used data from the Income and Perception Survey of the Unemployed Worker Support Program, conducted by the Social Foundation of the Chilean Chamber of Construction (CChC) (Cámara Chilena de la Construcción 2020) among 1724 unemployed construction workers. The survey was taken by telephone by social workers in late July 2021, as part of the social care framework belonging to the Unemployed Worker Support Program, financed by the CChC's social area. Crucially, this research is pioneering in measuring the impact of non-contributory cash transfers and pension fund withdrawals on the behavior of construction employment in Chile. Therefore, this study constitutes a first step in this line of research.

A main finding is that, if an unemployed construction worker withdrew some pension funds and benefitted from the EFI, the probability of not accepting a formal job offer increased on average 14%. This finding bolsters the hypothesis that pension fund withdrawals and government subsidies related to the health crisis have contributed to the shortage of formal labor in the construction sector. Based on estimations by the logit model, we found that the rejection probability increased with work experience (approximated by the worker's age). On the other hand, in general, informal work was not a sufficient condition to reject a formal employment offer. However, the probability of not accepting a formal employment contract increased 3.1% when the unemployed construction worker received income between US$317.50 and US$635 for informal work and, simultaneously, benefitted from the EFI and pension fund withdrawals.

Construction workers' income expectations is a relevant variable when accepting a formal proposal. For example, workers with expectations of rising incomes were less likely to reject a formal employment offer than those who expected incomes to fall or remain flat. Workers were willing to participate in the formal market in the hope that income will improve in the short term. Based on this result, it is worth analyzing if the average construction worker's reserve salary has increased. This claim is consistent with the results of the Business Perception Report of the Central Bank of Chile (Banco Central de Chile 2021b)—based on a survey of nearly 100 companies from different economic sectors. In particular, the executives surveyed highlighted a key reason for hiring difficulties was the demand for higher wages. So, this view (of those who demand labor) coincides with our findings from the perspective or workers' preferences (labor supply). In addition, our results provide a cause-and-effect relationship by sizing the impact of non-contributory cash transfers and pension fund withdrawals by households on their leisure preferences. We provide a scientific response to the problems construction companies experience while

trying to fill vacancies, and we contribute useful evidence for the design or calibrations of public policy to anticipate and respond to challenges of similar characteristics in the future (Atutxa et al. 2021).

The paper is organized as follows: Section 2 presents the cross-sectional data for survey and modeling methodology. The Section 2.2 outlines the parameterization of the logit model. Section 3 details the main results of the estimation of the model. In particular, the marginal effects of the determinants of rejection probability of a formal employment contract that were significant at 95% confidence are analyzed. Finally, Section 4 concludes.

## 2. Methodology

### 2.1. Data Collection

The database is cross-sectional and comes from the Income and Perception Survey of the Unemployed Worker Support Program, conducted by the CChC's Social Foundation among 1724 unemployed construction workers in Chile. The survey was carried out by telephone in late July 2021 by social workers of the Foundation in the context of social care and part of the Unemployed Worker Support Program, financed by the CChC's social area.

The survey involved 17 multiple choice questions, one of which was "*If you were offered a formal employment contract, would you accept it?*". The question served as the variable to be explained. This was a binary variable, allowing only a "yes" or "no" answer, but enabling in some cases the provision of reasons for rejecting an offer. For simplicity, we assigned value 1 to the variable if the respondent would not accept a formal employment contract and 0 if the answer was affirmative. On the other hand, we have considered 43 explanatory variables, grouped into the following categories: (i) characteristics of the unemployed construction worker (age, gender, education, among others); (ii) assets above the level of household financial expenses; (iii) location; (iv) income expectations; (v) engaging in informal work; (vi) pension fund withdrawals; and (vii) reception of state benefits or other subsidies such as the EFI. Table 1 provides descriptions of these variables.

**Table 1.** Definition of variables considered in the study.

| Variable | Definition |
|---|---|
| PFA withdrawal | The variable takes value 1 if the unemployed worker withdrew less than US$317.50 and value 0 in any other case. |
| Received EFI (less than US$317.50) | The variable takes value 1 if the unemployed worker received less than US$317.50 from the EFI and value 0 otherwise. |
| PFA withdrawal and EFI received | The variable takes value 1 if the unemployed worker withdrew PFA funds and received the EFI benefit. Value 0 otherwise. |
| Worked informally during April–June | The variable takes value 1 if the unemployed worker performed an informal job during the April–May–June quarter and value 0 otherwise. |
| Received between US$317.50 and US$635 for informal work during April–June | The variable takes value 1 if the unemployed worker received between US$317.50 and US$635 for performing informal work during the April–May–June quarter and value 0 otherwise. |
| Debt payments between US$317.50 and US$635 monthly | The variable takes value 1 if the unemployed worker declared a debt expense between US$317.50 and US$635 per month and value 0 otherwise. |
| Expects income increase within 12 months | The variable takes value 1 if the unemployed worker believed income would rise within the next 12 months and value 0 otherwise. |
| Gender | The variable takes value 1 if the unemployed worker was male and value 0 if female. |
| Age (proxy of work experience) | Continuous variable that measures the unemployed worker's age in years. |
| Has secondary or technical education | The variable takes value of 1 if the unemployed worker attended secondary or technical education. |

*2.2. Statistical Modelling*

To model and estimate the probability that an unemployed construction worker rejects a formal job proposal, conditional on determinants (43 explanatory variables constructed from the survey instrument of the Social Foundation), we used the logit model (Greene 2002; Trivedi 2009).

Let $y_i$ be the dependent variable or variable of interest. In the case that concerns us, $y_i$ is a dichotomous variable defined as

$$y_i = \begin{cases} 1, & \text{if the unemployed construction worker had rejected an} \\ & \text{offer for a formal employment contract;} \\ \\ 0, & \text{if such proposal was accepted.} \end{cases} \tag{1}$$

Formally, Equation (1) presents the parameterization of the qualitative response models (or probability models):

$$y_i = F(\mathbf{x}^\top \boldsymbol{\beta}) + \varepsilon_i, \quad \varepsilon_i \sim^{i.i.d} N(0, \sigma_\varepsilon^2), \tag{2}$$

where $\mathbf{x}$ is the vector of explanatory variables or variables that condition the probability of rejecting a formal job proposal; $\boldsymbol{\beta}$ is the vector of parameters to be estimated by maximum likelihood; $\varepsilon_i$ is the error component in the qualitative response model specification; and $F(\cdot)$ is the probability distribution function, such that:

$$F(\mathbf{x}^\top \boldsymbol{\beta}) = \frac{\exp(\mathbf{x}^\top \boldsymbol{\beta})}{1 + \exp(\mathbf{x}^\top \boldsymbol{\beta})}, \tag{3}$$

for a logit model.

The likelihood function to be maximized is:

$$L(\boldsymbol{\beta}) = \prod_{i=1}^{N} F(\mathbf{x}^\top \boldsymbol{\beta})^{y_i} (1 - F(\mathbf{x}^\top \boldsymbol{\beta}))^{1-y_i}, \tag{4}$$

where $N$ is the number of observations in the sample to be considered in the model (2). Finally, taking the logarithm of the previous function, we get the log-likelihood function to be maximized:

$$\mathscr{L}(\boldsymbol{\beta}) \equiv \log L(\boldsymbol{\beta}) = \sum_{i=1}^{N} y_i \log F(\mathbf{x}^\top \boldsymbol{\beta}) + (1 - y_i) \log (1 - F(\mathbf{x}^\top \boldsymbol{\beta})) \tag{5}$$

to obtain the estimator of parameter vector $\boldsymbol{\beta}$.

## 3. Results

Next, we highlight the data's main descriptive results. The survey showed that most respondents worked in the country's central macrozone. About 16% of the total sample were female workers, 40% said they completed secondary education, and the main occupations were at professional or day laborer levels (Figure 1). Meanwhile, 72% of respondents were aged 30 to 59. This is consistent with the years of experience considered in the present work and in the Construction Workers Characterization Report (Cámara Chilena de la Construcción 2019).

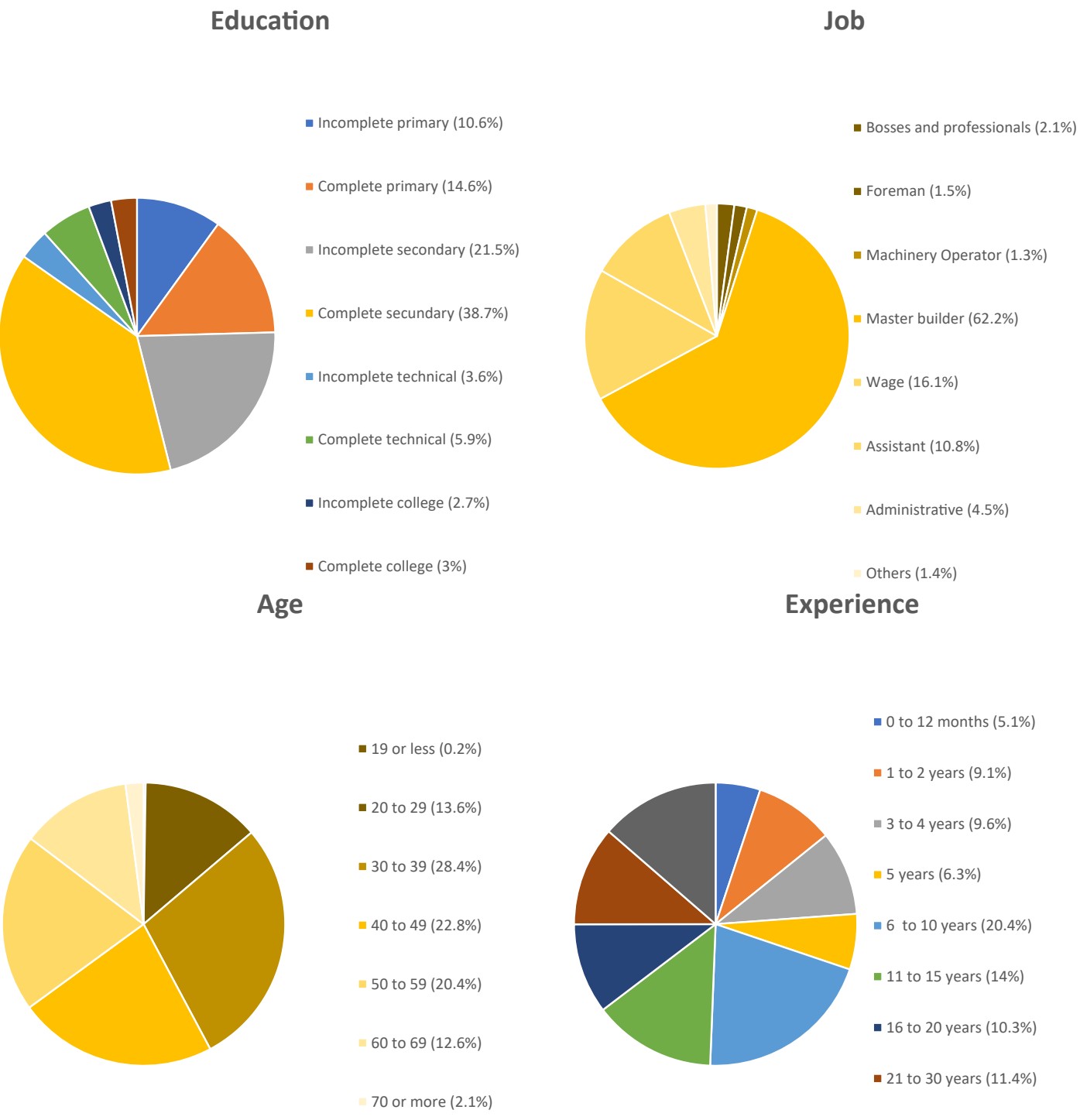

**Figure 1.** Pie charts of the variables education, occupation, age and experience of unemployed workers surveyed. Analogously to the US education system, primary and secondary education correspond to basic and secondary, respectively.

Workforce reductions figured among the most common reason for job loss, followed by finalizing a project, or lapsed contracts. And 73% of respondents sought formal work but were unsuccessful. Respondents that did not look for formal work (27%) were doing informal jobs or caring for family members. Several respondents rejected a formal offer because (i) the work or task had not yet started, (ii) the salary was insufficient, (iii) they had to care for family members, or (iv) they were not mobile, among others (Figure 2).

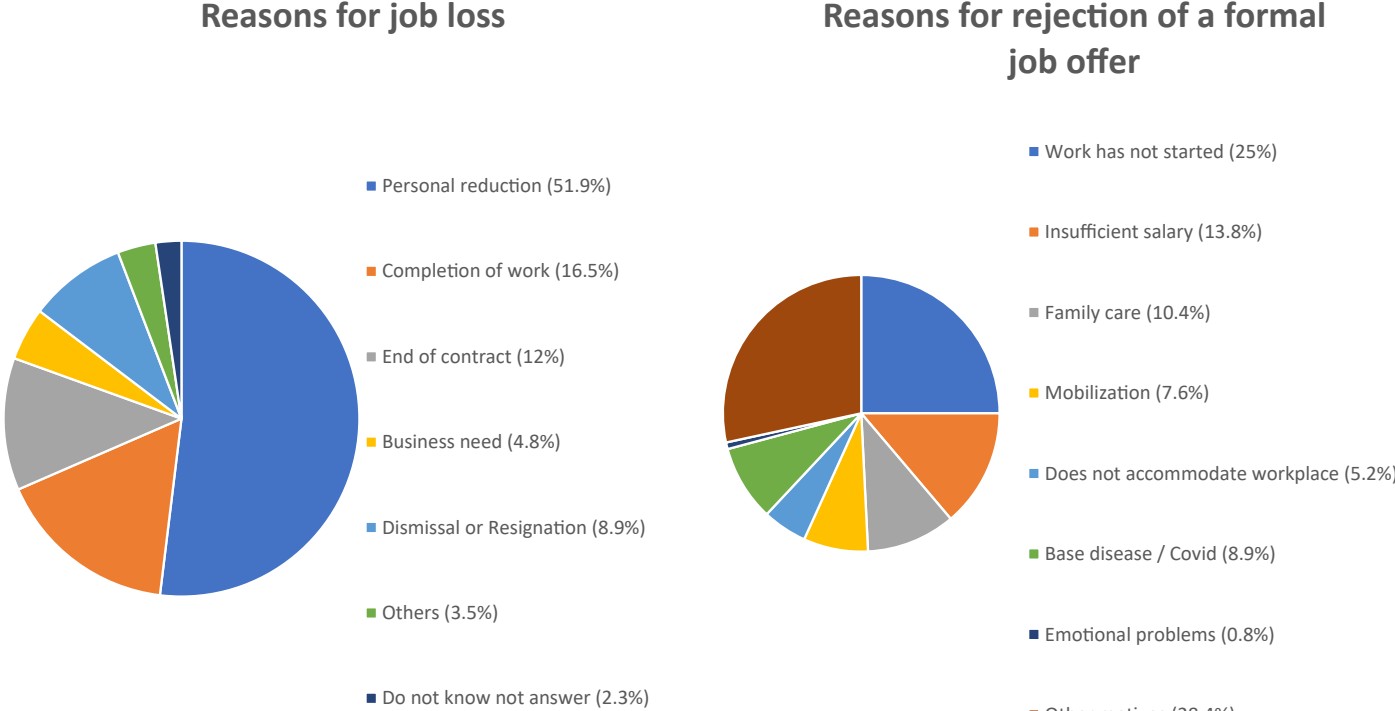

**Figure 2.** Pie charts on the variables that explain the reasons for losing a job and for rejecting a formal job offer. The results in the graph corresponding to the *reasons for rejection of a formal job offer* are derived from a subsample of 384 workers.

About 68% of respondents did informal jobs during the second quarter of 2021 (Table 2), with 38% receiving a salary of less than US$317.50 and 26% receiving between US$317.50 and US$635. Also, 39% withdrew pension funds, while around 86% had received the EFI. Only 7% said they saved their severance pay, 5% received another type of subsidy/income, and 13% received funds from their unemployment insurance. Table 2 shows that 95% of unemployed workers spent at least US$635 on services and food, while 98% declared similar expenses associated with debt (credits, loans, mortgages, leases, among others). Therefore, under the premise that these proportions can be extrapolated to the population of unemployed workers in the construction sector, the main results are:

- Of the total, 68% would work in the informal market. Of these, 64% claimed to have received less than US$635 per month. This amount is not substantially different from a construction worker's average market salary of US$676.91 per month.
- Also, 72% of respondents had debt of less than US$317.50. Assuming most of these credits were acquired in the formal financial market, the amount could count as debt deferral benefit, involving an extension of loan maturity implemented in 2020 by the Financial Market Commission (FMC), which regulates the sector, and the Central Bank.
- Considering informal income and the debt deferral benefit, a percentage of unemployed workers could be receiving an amount higher than the average market salary (around US$676.91 per month). Therefore, from a worker's short-term perspective, working in the informal market carries greater benefit than cost.

**Table 2.** Distribution (in percentage) of income from informal work and monthly expenses of unemployed workers.

| Factor | No Applies | Less than US$317.50 | From US$317.50 to US$635 | From US$635 to US$952.50 |
|---|---|---|---|---|
| Informal work | 32 | 37.9 | 26.2 | 3.9 |
| Basic services and food | 0 | 38.7 | 56.5 | 4.4 |
| Debts | 0 | 72.1 | 25.7 | 2.2 |

Next, we analyze the results for each of these explanatory variables, obtained from programming in MATLAB code (Matlab 2019). Table 3 shows that if an unemployed construction worker withdrew some pension funds and benefited from the EFI subsidy, the probability of not accepting a formal job offer increased on average 14%, with an interval of 6% to 21% at 95% confidence level. Empirical evidence, based on data collected by the CChC's Social Foundation, favors the hypothesis that pension fund withdrawals and government subsidies during the health crisis have, to some extent, contributed to the shortage of formal construction labor. However, if pension fund withdrawals were below US$317.50, the probability of rejection decreased by 9%, as shown in Table 3. Likewise, the more the EFI subsidy fell below US$317.50, the lower the rejection probability. These results are consistent with the above-mentioned hypothesis.

**Table 3.** Main results of the estimation of logit model. *N* are the number of observations in which the worker declared to be over 80 and 90 years of age were eliminated (1% of the sample). * Significative value at 1% of confidence level. ** Significative value at 5% of confidence level.

| Variable | Coefficients | | | Marginal Effect | | |
|---|---|---|---|---|---|---|
| | Value | St. Error | *p*-Value | Value | St. Error | *p*-Value |
| Constant | −1.893 (*) | 0.434 | <0.01 | −0.232 (*) | 0.052 | <0.01 |
| PFA withdrawal | −0.719 (*) | 0.272 | <0.01 | −0.088 (*) | 0.033 | <0.01 |
| Received EFI (less than US$317.50) | −0.309 (**) | 0.141 | 0.028 | −0.038 (**) | 0.017 | 0.028 |
| PFA withdrawal and EFI received | 1.097 (*) | 0.331 | <0.01 | 0.135 (*) | 0.040 | <0.01 |
| Worked informally during April–June | −1.320 (*) | 0.168 | <0.01 | −0.162 (*) | 0.020 | <0.01 |
| US$317.50–US$635 received for informal work during April–June | −0.845 (*) | 0.184 | <0.01 | −0.104 (*) | 0.023 | <0.01 |
| Between US$317.50 and US$635 monthly debt expense | −0.393 (**) | 0.168 | 0.020 | −0.048 (**) | 0.021 | 0.019 |
| Expects rising incomes within 12 months | −0.472 (*) | 0.146 | <0.01 | −0.058 (*) | 0.018 | <0.01 |
| Gender | −0.531 (*) | 0.170 | <0.01 | −0.065 (*) | 0.021 | <0.01 |
| Age (proxy work experience) | 0.023 (*) | 0.006 | <0.01 | 0.003 (*) | 0.001 | <0.01 |
| Declared a level of secondary or technical education | −0.303 (**) | 0.144 | 0.036 | −0.037 (**) | 0.018 | 0.036 |
| *N* | | | 1704 | | | |
| Wald test | | | 536.825 (*) | | | |
| *p*-Value | | | <0.01 | | | |
| Predictive capacity assessment: | | | | | | |
| Accuracy when observed variable = 1 | | | 61% | | | |
| Accuracy when observed variable = 0 | | | 84% | | | |

Another finding that emerges from Table 3 is that working in the informal market did not necessarily lead to rejecting a formal job proposal. This is because the sign of the coefficient that accompanies the variables related to informal work is negative. Therefore, this condition is not sufficient to link to the non-acceptance of a formal job offer. However, if an unemployed worker received between US$317.50 and US$635 for informal work and, at the same time, benefitted from the EFI and pension fund withdrawals, formal employment rejection probability increased by 3.1%. The debt situation was relevant when deciding on a formal job proposal. If debt payments were US$317.50 to US$635 per month, the probability of rejecting a formal job offer decreased by 5%.

Workers expecting rising incomes during the next 12 months would more likely accept a proposal for formal employment compared to those who expected incomes to decrease or remain flat. This result indicates that on average unemployed construction workers expected to receive higher remuneration in the short term, which could imply an increase in the reserve salary. This finding is consistent with the results of the Business Perception Report of the Central Bank of Chile (Banco Central de Chile 2021b)—based on a survey of around 100 companies from different economic sectors. In particular, 80% of respondents said they encountered difficulties in filling vacancies during 2021. They said one of the most prominent reasons for these difficulties was high salary demands by candidates. Therefore, this view from the labor demand side is consistent with our findings on the labor supply side. Besides, the significant coefficient with a negative sign that accompanies the gender variable indicates that women more likely reject a formal job offer than men.

This could be mainly due to women's greater commitment to caring for minors, disabled family members or an older adult member of the household.

Age is a variable commonly correlated with years of work experience. In this case, the worker's age and the experience variable (range greater than 10 years) have a positive correlation of 0.52. However, the "age" variable, being continuous, enabled fine-tuning marginal effects with respect to a variable measured in ranges of years. This way, it can be concluded that the more work experience, the greater the probability of rejecting a formal employment contract. This category of workers likely demanded a higher salary than the current average salary offered by construction companies. Thus, as age (proxy variable of experience) increased, formal contract rejection probability increased by 0.3%. The higher the education level, the lower the probability of rejecting a formal employment contract. In effect, rejection probability decreased by 3.7% when the unemployed worker had secondary or technical education.

Finally, Figure 3 shows that the probability of rejecting a formal contract increased with years of experience (approximated by age). Likewise, a worker with a certain level of work experience, who also withdrew pension funds and received EFI, was significantly more likely to reject a proposal for a formal employment contract than a worker with the same experience, but without receiving these benefits. In addition, Figure 3 shows that the more experience/age an unemployed construction worker had, who also received PFA and EFI benefits, the higher the probability of not accepting a formal job offer. This could imply an increase in reserve salary; mainly for workers who have withdrawn pension funds and received a state subsidy.

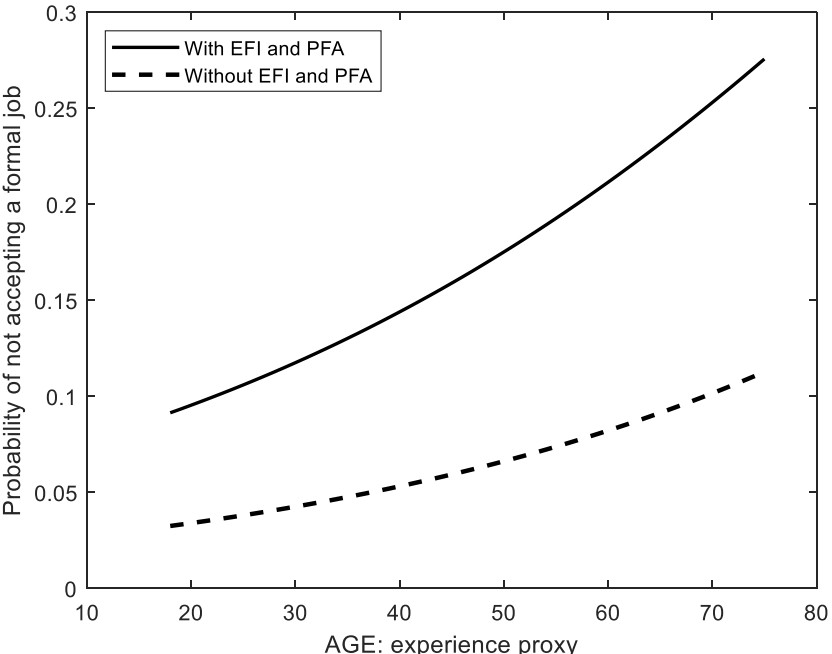

**Figure 3.** EFI and PFA withdrawal effect on the probability of not accepting a formal job.

## 4. Conclusions

Based on observations from the Income and Perception Survey of the Unemployed Worker Support Program of the CChC's Social Foundation, conducted among 1724 unemployed construction workers in Chile, we found statistical evidence that the withdrawal of pension funds and the EFI together increased the probability that an unemployed construction worker will reject a formal contract proposal. This finding favors the hypothesis that pension fund withdrawals and government subsidies related to the health crisis have, to some extent, contributed to the shortage of formal construction labor. Rejection probability increased with work experience (approximated by worker age), according to our findings. In this sense, income claims for this category of construction workers have increased on average, due to higher transfers from the state and other benefits or household subsidies.

In this regard, Box 1b in (Cámara Chilena de la Construcción 2021) shows some indications that scarce construction labor, among other factors, is due to an increase in workers' reserve wage (minimum wage for which a worker would be willing to accept a formal employment contract). The higher reserve salary of workers is due to the higher household income from government transfers, among other benefits that have generated a wealth effect, including pension fund withdrawals and postponement of financial obligations (mortgage loans and/or payments for basic services). To this added income from informal work or self-employment, due to the growing demand for home repairs or remodeling by households—financed by a fraction of pension savings withdrawn to improve conditions in a teleworking context. The latter is consistent with Central Bank findings in (Banco Central de Chile 2021a).

The present findings constitute the first empirical evidence that the withdrawal of pension funds and the EFI have contributed to the shortage of formal labor in the construction sector and to be considered for policy decision-making (Smart and Burgos 2018). In quantitative terms, the probability of not accepting a formal contract by a worker with high experience, who also withdrew PFA funds and received the EFI, increased 28%. This is about 2.5 times the rejection probability of a worker with the same experience, but without having received additional PFA and EFI income.

This way, our research provides a cause-and-effect relationship by sizing the impact of non-contributory cash transfers and pension fund withdrawals by households on their work preferences. We have provided a scientific response to the problems construction companies currently experience while trying to fill vacancies. We contribute useful evidence to anticipate and respond to challenges of similar characteristics in the future, understanding that the COVID-19 pandemic is a dynamic phenomenon that unevenly affects different economic sectors (Atutxa et al. 2021). The construction sector has been among the most affected because it requires intense on-site labor.

**Author Contributions:** B.J.I.-A. and J.E.C.-R. wrote the paper and contributed reagent/analysis/material tools; B.J.I.-A. conceived, designed, and performed the experiments and analyzed the data. All authors have read and agreed to the published version of the manuscript.

**Funding:** Research was fully supported by FONDECYT (Chile) grant No. 11190116.

**Institutional Review Board Statement:** Not applicable.

**Informed Consent Statement:** Not applicable.

**Data Availability Statement:** Restrictions apply to the availability of these data. Data was obtained from the Income and Perception Survey of the Unemployed Worker Support Program and are available from the authors with the permission of the CChC's Social Foundation.

**Acknowledgments:** The authors thank the editor and two anonymous referees for their helpful comments and suggestions. The authors also acknowledge the valuable collaboration of the CChC's Social Foundation is appreciated for sharing its unnamed database of the Income and Perception Survey of the Unemployed Worker Support Program. The support Camilo Torres in data processing is especially appreciated.

**Conflicts of Interest:** The authors declare no conflict of interest.

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
