# Peer review of "Monetary Fiscal Contributions to Households and Pension Fund Withdrawals during the COVID-19 Pandemic: An Approximation of Their Impact on Construction Labor Supply in Chile"

_socsci, doi:10.3390/socsci10110417_

Round 1

Reviewer 1 Report

This study shows the impact of COVID 19 on unemployment drawing example from mandatory private Pension Fund Administration.  It is a timely topic which has great importance. 

However, the following must be done to improve on the quality of the paper 

  • data should be integrated into methodology
  • education, job, age and experience to be represented under result specifically made under the sub-section of description of sample 
  • the author(s) should opt for either logit or probit, but not both since they are similar 
  • Table 3, author (s) should include P-value. The author (s) should also include standard errors, instead of standard deviation 

Author Response

Dear anonymous reviewer:

I would like acknowledge this careful revision of our manuscript socsci-1412175 titled: "Monetary fiscal contributions to households and pension fund withdrawals during the COVID-19 pandemic: An approximation of their impact on construction labor supply in Chile". I am grateful that this manuscript can be considered for publication after mayor revision. We also thank the reviewer for all their valuable comments and constructive criticism. I have included (see attached file below), a detailed point-by-point response to all the reviewer's comments and suggestions. The comments from the reviewer are listed in cursive italic letters, and updated lines appear in red in the manuscript.

With kind regards,
Corresponding author

Reviewer 2 Report

Thanks for submitting this paper. I found that the paper is interesting and the topic is related to recent pandemic. 

While it is just a brief report, I think in the introduction and conclusion the authors can try to compare with the past studies.

Author Response

(The authors gave the same response as above.)

Round 2

Reviewer 1 Report

The author(s) have improved on the quality of the work as compared from the previous submission. However, the following issues have to be addressed clearly. 

  • the authors are advised to use logit only into the model. So leave out probit model.
  • Table 3 - both coefficient and marginal effect have their own P-values and standards errors. Against each coefficient and marginal effect indicate P-value with asterick. For example, means significant at 1 %   significant at 5%. 

Author Response

Dear Reviewer:

I would like acknowledge this careful revision of our manuscript socsci-1412175 (3rd version) titled: "Monetary fisscal contributions to households and pension fund withdrawals during the COVID-19 pandemic: An approximation of their impact on construction labor supply in Chile". I am grateful that this manuscript can be considered for publication after minor revision. We also thank the reviewer for all their valuable comments and constructive criticism. I have included (see attached file below), a detailed point-by-point response to all the reviewer's comments and suggestions. 

The author(s) have improved on the quality of the work as compared from the previous submission. However, the following issues have to be addressed clearly:

1. the authors are advised to use logit only into the model. So leave out probit model.

R: According to your suggestion, we omited probit model in all parts of manuscript.

2. Table 3 - both coefficient and marginal effect have their own P-values and standards errors. Against each coefficient and marginal effect indicate P-value with asterick. For example,means significant at 1 %   significant at 5%. 

R: According to your suggestion, now the table includes the estimations and p-values for coefficients and marginal effects, and asterick and double asterick appears in parenthesis to indicate significance at 1% and 5%, respectively. Table 3 and 4 were merged in this new version.

With kind regards,
